# Analysis of Correlation between Structure and Properties of Carboxymethyl Cellulose Film Loaded with Eu^3+^ and Tb^3+^ Fluorescence by Rheology at Different Drying Stages

**DOI:** 10.3390/polym14091655

**Published:** 2022-04-20

**Authors:** Jun Ye, Zichang Fu, Jiawei Rao, Jian Xiong

**Affiliations:** 1School of Light Industry and Engineering, South China University of Technology, Guangzhou 510640, China; jye@scut.edu.cn (J.Y.); fzcyouxiang@hotmail.com (Z.F.); jasperrao@outlook.com (J.R.); 2College of Light Industry and Food Engineering, Guangxi University, Nanning 530004, China; 3School of Food Science and Engineering, South China University of Technology, Guangzhou 510640, China

**Keywords:** fluorescent carboxymethyl cellulose, rheological properties, mechanical property, polymer chain

## Abstract

The influences of interactions between carboxymethyl cellulose (CMC) and CMC/europium (III)–terbium (III) (CET) on the structure and properties of the resultant CMC/CET films were investigated by rheology at three stages of the film-drying process. According to the water content at different drying times, the kinetics curves during the film-drying process were drawn. Then, the rheology properties of film-forming solutions during the drying process were characterized by dynamic modulus, Han plots, zero shear complex viscosity and relaxation time. When the water content was 90%, the film contained either 0.1 or 0.2 g of CET, which had good fluidity, while the film with 0.3 g of CET was elastic-dominated. Han plots and XRD analyses showed that the interactions between the CMC and CET were not hydrogen bonds but random entanglements. The zero-shear complex viscosity and relaxation time spectrum confirmed that the entanglements enhanced as the CET content increased. Meanwhile, aggregation formed in the solution of CMC with 0.3 g of CET. When CMC-CET films with different CET additions were compared, the film with 0.2 g of CET had an even and tight sheet structure, the greatest fluorescence intensity, and superior tensile strength of 78.76 MPa.

## 1. Introduction

Carboxymethyl cellulose (CMC), as a good film-forming material, has excellent biodegradation and bioactivity and has been extensively sought after for numerous applications. For example, Noshirvani et al. prepared a new nanocomposite film as a packaging material by incorporating chitosan-CMC-oleic acid with ZnO nanoparticles, which inhibited the growth of fungi and improved the shelf life of food [1]. Likewise, Saadiah et al. prepared a composite film using CMC/PVA hybrid polymers to serve as the host of polymer electrolytes, which has increased the ion conductivity by one magnitude order from 10^−7^ to 10^−6^ S/cm [2]. Ye et al. prepared fluorescent paper sheets by coating CMC/Eu(III) or CMC/Tb(III) onto a cellulosic paper, which offered a simple and effective route to prepare quality fluorescent papers in the paper-making industry [3]. Bowers et al. designed hyaluronic acid/CMC films as a barrier to prevent adhesions and evaluated the effects on perianastomotic adhesions in previously irradiated rats [4].

Drying processes, such as heating, can affect the quality of the films during their formation [5,6,7,8,9,10,11]. For example, Gregorova et al. revealed that increasing the drying temperature can initiate the interaction between CMC and PVP, resulting in an increase in stiffness by physical cross-linking [12]. With an NMR scanner, Ghoshal et al. followed gelatin from porcine skin solutions dissolved in D_2_O during the drying process until complete solidification occurred [13]. They also investigated the progress of molecular mobility of starch suspension until the final stage of film drying by spin–spin relaxation times, which revealed that the network formation and precipitation resulted in an increased crystallinity at the bottom surface [14]. Perez-Gago and Krochta reported that drying temperature increased the lipid crystalline morphology and lipid distribution within the whey protein isolate (WPI) matrix [15]. In the meantime, the water vapor permeability decreased for WPI-lipid films as compared to the WPI films. Additionally, during the drying process, there might be a phase transformation. For example, Denavi et al. reported that the thermal gelation transited from a rubbery to a vitreous phase. Nevertheless, some of these reported methods were costly and needed a tedious operation and long test cycle [16]. More importantly, although these studies have suggested how films structures impact their properties, the fundamental understanding of the structure–function relationship is still not well understood.

As the heating and drying process proceeded, evaporation of the solvent changed the rheology of the film-forming solution. Rheological properties were sensitive to the film formation process, so most researchers focused on investigating the rheology of the initial film-forming solution and the performance of films. Chun and Ko [17], La Mantia et al. [18], Münstedt et al. [19], Yusova and Lipatova [20], El Miri et al. [21] studied the rheology of film-forming solutions and tensile properties of CMC/starch films reinforced with cellulose nanocrystals. They affirmed a transition from Newtonian behavior to shear thinning, which occurred when cellulose nanocrystals were added. Silva-Weiss et al. reported that viscosities of the film-forming solutions of chitosan and chitosan-corn starch with the extract from murta leaves increased with the addition of the extract, leading to a gel-like structure and thixotropic behavior [22]. Löfflath and Gebhard examined the rheological changes during the initial drying stages of a water-based latex film at different pH, with different co-solvents and neutralizing agents [23]. They found that water and co-solvent evaporation rates were determined as a function of temperature and humidity. However, they did not show the results regarding the film structure and property.

In this research, samples with different water contents during different drying times were selected. The interactions between the CMC and CET of the samples at the different drying stages, such as crystallization, entanglement and chain conformations of macromolecules, were then investigated by rheology properties during the process of film-forming, such as dynamic modulus, complex viscosity, Ham plot and Maxwell model on relaxation spectra. Our objective of this research could be, finally, to reveal the relationship between the structure and properties of the blend films by the rheology properties during the process of film-forming.

## 2. Materials and Methods

### 2.1. Materials

CMC (degree of substitution DS = 0.89, food grade, number-average molar mass Mn = 2.17 × 10^5^ by GPC with wide angle laser light scattering photometer) was produced by Yingte Chemical (Shijiazhuang, Hebei Province, China) Co., Ltd.; Eu_2_O_3_ (AR grade), TbCl_3_·6H_2_O (AR grade) was produced by Aladdin Chemical Reagent (Shanghai, China) Co., Ltd.; NaOH (AR grade) and Guangzhou Donghong Chemical Plant (Guangzhou, Guangdong Province, China), respectively; HCl (AR grade) and ethanol (AR grade) were purchased from Guangzhou Chemical Reagent Factory (Guangzhou, Guangdong Province, China); KBr (spectral pure) was produced by Tianjin Komio Chemical Reagent (Tianjing, China) Co., Ltd.; Dialysis membrane (MWCO: 2000), Shanghai Yuanye Biotechnology (Shanghai, China) Co., Ltd. Products. All reagents were used without further purification.

### 2.2. Methods

#### 2.2.1. Preparation of CET Composite

CET was prepared as reported by Ye et al. [24]. In our experiments, the amount of CMC was 1.000 g. The concentrations of both TbCl_3_ and EuCl_3_ solutions were 0.0318 mol/L, and the amount of both solutions was 15 mL. The reaction was carried at pH = 7.00 and 70 °C for 30 min with stirring by a magnetic stirrer. Then, the mixture was dialyzed with deionized water until there was no AgCl precipitation when AgNO3 was added, and then the mixture was dried in an oven at 70 °C. The structure of CET is shown in Figure 1 [24], which displayed that CMC connected with Tb^3+^ and Eu^3+^ both by inonic and cooerdinate bonds.

#### 2.2.2. Preparation of Blend Films

CMC was dissolved in 30 mL of deionized water. The CET sample was added according to Table 1 and stirring was continued. The prepared S0, S1, S2, S3 film-forming solutions (see Table 1) were cast onto a mold of 10 cm × 10 cm and cured at 50 °C, respectively. All of the films were stored in a desiccator until use.

In Figure 1, a, b, c, d, e, f indicate the structure of CMC, the structure of CET, the film-forming solution, the film, the AMF micrograph of the blend film, the SEM image of cross section of the blend film, respectively.

#### 2.2.3. Characterization

S0, S1, S2, S3 film-forming solutions (see Table 1) were poured into Petri dishes and dried in an oven at 50 °C. Water loss was periodically weighed by taking out the Petri dishes and weighing them on the electronic balance with a precision of 0.0001 g. The end point of drying was taken to be when no further changes in weight were observed. All weighing processes were completed in less than 10 s during the drying process. According to the data of time and weight, the drying kinetics curves were drawn; rheological properties were measured with an ARES-G2 rheometer equipped with a parallel plate (60 mm diameter; 500 μm gap). A dynamic frequency sweep was conducted by applying an oscillation amplitude within the linear viscoelastic region (5% strain) over a frequency range between 0.1 and 100 Hz at 50 °C. The relaxation spectra were obtained by TRIOS software. The crystallization performance was measured by a D8 Advance X-ray diffractometer manufactured by Bruker, Germany. The Cu target, Kα ray, tube pressure was 40 kV, tube flow was 40 mA, and the scanning range was 5° to 50°. The AFM test was performed by a Multimode IIIA atomic force microscope, manufactured by Veeco, USA. The probe material model was RTESP, the cantilever elastic constant was 40 N/m, the scanning speed was 1 Hz, and the tapping mode was used to measure at room temperature. Morphology analysis was carried out using an EVO18 scanning electron microscope from Carl Zeiss, Germany. The PL test was carried out by Fluorolog-3 fluorescence spectrometer produced by the JY Company of the United States. The fluorescence excitation spectrum was measured at a wavelength of 545 nm, and the fluorescence emission spectrum was measured at an excitation light of 374 nm. The number-average molecular weight (Mn) of CMC was determined by GPC on a WATERS 515 equipped with wide angle laser light scattering photometer (DAWN HELEOS) (laser wavelength is 658.0 nm) and laser differential refractive photometer (Optilab rEX). The mechanical testing was measured using the INSTRON 5565 tensile and compression material testing machine produced in the United States. The film was cut into a rectangular strip of 10 cm × 10 cm. The initial setting was 50 mm, and the stretching speed was set to 1 mm/s. The temperature is 24.54 °C, and the humidity is about 43.2% RH; the variance analysis was performed on the data using R software, *p* < 0.05.

## 3. Results and Discussion

### 3.1. Dynamic Rheological Properties of Film Forming Solutions

#### 3.1.1. Drying Kinetics Curves of Film-Forming Solutions

Blend film-forming solutions with initial water contents in the range of 97.72–98.68 w% were dried until their equilibrium moisture contents, which was to construct the drying kinetics curves. The drying kinetics curves of all of the film-forming solutions during the drying processes are shown in Figure 1.

It was also observed that all curves had three stages during the drying process. The same result was described as the latex film-forming solution by Löfflath and Gebhard [23]. In the first stage, the rates of water evaporation of all samples were constant. Water contents of all film-forming solutions decreased linearly and evenly. When the water contents of the film-forming solutions reached 90%, all curves turned to the second stage, where the rates of water evaporation of all samples decreased dramatically. The water contents of these film-forming solutions decreased sharply from 90% to 10%. After that, these curves turned to the third stage, where the film formed. Finally, the water contents of these films were kept constant at about 8%. The content of CET has little effect on the film formation time, indicating that CET did not destroy the combination of CMC and water. In the first stage, all of the film-forming solutions were still in a fluid state. Notably, in the second stage, all of the film-forming solutions developed solid-like behavior (yield value). The yield value increased until the films were essentially a solid with the water contents of these film-forming solutions at 10%. The third stage involved the slow diffusion of water out of the films.

#### 3.1.2. Dynamic Modulus Characterization of the Film-Forming Solutions during the Drying Processes

Figure 2a–d shows the loss moduli (G″) and storage moduli (G′) of the initial film-forming solutions. Within the tested frequency range (1 to 100 rad/s), the G″s of all initial film-forming solutions were higher than the corresponding G’s of these initial film-forming solutions, with a tendency to approach each other at high frequencies (around 100 rad/s).

The dynamic moduli (G′and G″) of all film-forming solutions with water content of 90% are presented in Figure 2e–h. Figure 2e–h clearly indicate that the rheological properties of the film-forming solutions during the drying processes were significantly different. For S1-90, liquid-like viscoelastic behavior was still observed (Figure 2f). The rheological behavior of S2-90 was similar to that of CMC, meaning their G’s increased more rapidly with frequency than G″s.

When the film-forming solutions dried to a water content of 30%, dynamic spectra of S1-30 and S2-30 at a given frequency exhibited a significant difference (Figure 2i–l). The G″ of S1-30 was higher than G´ at low frequencies (Figure 2j). Notably, the G’s of S2-30 and S3-30 were significantly higher than their G″s from 1 to 100 rad/s (Figure 2k,l).

It can be seen from the dynamic modulus information that these initial film-forming solutions with CET exhibited typical liquid-like viscoelastic behavior such as a CMC solution.

When the film-forming solutions dried to a water content of 90%, the curves of S0-90 intersected at about 10 rad/s, and G´ of S2-90 was almost equal to its G″ in the frequency range 1–10 rad/s, which means that the liquid–solid transition happened for S0-90 and S2-90 [25]. Moreover, the interactions of the polymers in S2-90 were stronger than that in S0-90. G´ of S3-90 was higher than G″, showing obvious elastic behavior (see Figure 2h). It was worth noting that, from S1-90 to S3-90, as CET water content decreased, the frequency at which G″ almost intercepted G´ was lower. These results indicated that CET contributed to a larger extent to elasticity at this stage of the drying process. This might be attributed to the increase in inter-chain couplings between CET and CMC.

When the film-forming solutions dried to a water content of 30%, the curves of S1-30 intersected at 40.5 rad/s, showing a typical concentrated polymer solution [26]. At this time, entanglements between chain segments of CMC and chain segments of unbonded CET were observed in S1-30. G’s of S2-30 and S3-30 were significantly higher than their G″s, indicating that these systems were predominantly elastic. Furthermore, G´ and G″ were almost parallel to each other and slightly frequency-dependent. This might be attributed to weak gel formation after inter-chain entanglements between chain segments of CMC and chain segments of unbonded CET at this drying stage [27].

#### 3.1.3. Han plots of the Film-Forming Solutions during the Drying Processes

The Han plot is to reveal the lgG′–lgG″ relationship, which has been widely used to judge the compatibility and phase separation of the blend system of Hao et al. [28,29]. If the relationship matches Equation (1), there is less phase separation between the polymers in the film-forming solution.
(1)$$\begin{array}{c} lgG\prime \;\propto \;2lgG\prime \prime \; \end{array}$$

Figure 3 shows the Han plots of the film-forming solutions dried to different water content and the XRD diffraction pattern of the S0–S3 film. As shown in Figure 3, for the initial film-forming solutions, the Han plots of the four samples overlapped, suggesting that there were some interactions between the CMC chain and the unbonded chain segment in the CET macromolecule. [30]. At the low-frequency region, the slope of the Han plots was close to 2, which indicated good compatibility between CET and CMC for all of the initial film-forming solutions.

The Han plots of S0-30, S0-90, and S1-30, S1-90 still overlapped well, respectively (Figure 3), indicating that they still had good compatibility. For S2-90, the slope of the Han plot was close to 2. However, the slope of the Han plot of S3-90 deviated from 2. This result indicated that there was phase separation in the S3-90 [28]. Moreover, this phenomenon was more significant in the S2-30 and S3-30. The phase separation contributed to different chain structures between CMC and CET in which CMC was cross-linked by Eu^3+^ ions or Tb^3+^ ions.

The XRD pattern of the S0–S3 film is shown in Figure 3d. The crystallinity can be obtained by the following formula [31]:(2)$$Cr.I_{.}=\left(I_{002}-I_{am}\right)/I_{002}\times 100\%$$
where *Cr.I.* is the degree of crystallinity, *I*_002_ is the maximum intensity of the (002) lattice diffraction and I_am_ is the intensity diffraction at 2θ = 18°.

The crystallinity of the S0–S3 film was calculated to be 23.1%, 23.0%, 18.6%, and 15.8%, respectively. Obviously, a small amount of CET macromolecules in the S1 film just entangled with the chains of the CMC in amorphous rather than destroyed the order of the CMC chains. Increasing the addition of CET led to an increase in the interactions of CMC chains with CET in crystalline during the film-forming solutions, resulting in a decrease in crystallinities of other films. This indicated that the interaction is not a hydrogen bond but a random entanglement during the drying process.

#### 3.1.4. Complex Viscosity of the Film-Forming Solutions during the Drying Processes

The complex viscosity (η*) of different film-forming solutions dried to different water content was plotted against the frequency, as shown in Figure 4. η* values at a given frequency increased significantly as drying progressed. Moreover, for all initial film-forming solutions, a Newtonian plateau occurred only at lower frequencies and shear thinning was observed at higher frequencies. For dried film-forming solutions, the Newtonian plateau disappeared, and shear thinning was observed over the tested range.

In Table 2, n is the nth sample of the different film-forming solutions, while I, 90 and 30 represent the initial water content, 90% water content and 30% water content of the film-forming solutions, respectively.

The zero-shear complex viscosity ($\eta_{0}^{*}$
) of each sample calculated according to the Cross model is shown in Table 2 [32]. As mentioned above, the addition of CET resulted in entanglement between the CMC chain and the unbonded chain segment in CET. However, the addition of CET weakened the interaction between the CMC chains. The addition of CET made the distance between the CMC chains increase, which greatly weakened the interactions between the CMC chains. This result was similar to the CMC solution diluted by CET; thus, we named the result the “dilution effect”.

The $\eta_{0}^{*}$ of the S1-I, S1-90 and S1-30 was less than that of S0-I, S0-90 and S0-30, respectively. Despite the good compatibility, the tiny content of CET may lead to less entanglement in the S1 film-forming solution. Once the entanglement was weaker than the “dilution effect”, a decrease in $\eta_{0}^{*}$ occurred.

The $\eta_{0}^{*}$ of S2-I was less than that of S0-I, while during drying, $\eta_{0}^{*}$s of S2-90 and S2-30 were higher than that of S0-90 and S0-30, respectively. This result indicated that CET can “dilute” the CMC solution for the S2-I, and the entanglement increased as the water content decreased. The interactions of entanglements took over major roles as drying processes, which resulted in an increase in the $\eta_{0}^{*}$s [33]. For the S3 film-forming solution, all samples had larger $\eta_{0}^{*}$ than that of S0, indicating that more entanglement existed in the S0 film.

#### 3.1.5. Study on Relaxation Spectra of the Film-Forming Solutions during the Drying Processes

The viscoelastic behavior of the film-forming solutions can be analyzed using the generalized Maxwell model, and the expression of the relaxation modulus is as follows:(3)$$G\left(t\right)=\overset{N}{\underset{i=1}{\sum }}G_{i}\exp\left(-t/\lambda_{i}\right)$$
where *G_i_* and *λ_i_* are the relaxation moduli and relaxation time of the ith element of the total number *N* of the modes [34].

Figure 5 shows the relaxation time spectrum of all film-forming solutions dried to different water contents. As shown in Figure 5, the relaxation modulus increased gradually with the decreasing water content of film-forming solutions. As the CET content increased, the relaxation modulus of film-forming solutions shifted to higher values in the long-range relaxation area due to the increased influence of the dissolved CET chain segment, which has slower relaxing modes.

If the film-forming solution is regarded as composed of multiple parts, each part can be regarded as a Maxwell unit. With that being said, the sum of the contributions of each Maxwell unit to the relaxation modulus is the relaxation modulus of the film-forming solution [34].

The relaxation tendency of S0-I, S0-90, S0-30, S1-I, S1-90, S1-30, S2-I and S3-I was found to be similar in that the relaxation modulus decreased with increased relaxation time, while the relaxation tendency of S2-90, S2-30, S3-90 and S3-30 changed. As compared to S3-30, the relaxation modulus of S2-30 changed very gently. Moreover, the contributions by each Maxwell unit to the modulus of S2-30 were consistent. However, there was a significant decrease in the modulus of S3-30 in the small-scale relaxation region. Especially, the contribution of each Maxwell unit to the modulus is largely different. This suggested that the distributions of the interactions between of CMC and CET macromolecules of S3-30 were more nonuniform than that in the S2-30. As the CET content increased, CET macromolecules aggregated severely and nonuniformly, but some of the unbonded chain segments in CET did not entangle with CMC chains. Thus, great changes in the contributions of each Maxwell unit to the modulus may infer the nonuniform distribution of interactions.

From the rheology research, we confirmed that there were entanglements rather than crystallization between the CMC chain and the unbonded chain segment in CET macromolecule during the drying process, but the interactions were nonuniform at higher CET contents. The mechanism of interactions between the CMC chain and the unbonded chain segment in the CET macromolecule during the drying process is shown in Figure 6.

### 3.2. Relationship between the Structure and Properties of Blend Films and the Rheology of Film-Forming Solution

#### 3.2.1. The Relationship between the Morphological Structure of Blend Films and the Rheology of the Film-Forming Solution

The AFM morphology of the S0–S3 films is shown in Figure 7. The average roughness (R_a_) of the films was 1.42, 3.79, 5.03, 5.60 nm, respectively, showing that the surface of the blend film was rougher. When the amount of CET added was 0.3 g, the roughness of the film reached the highest value.

Compared with R_a_ of S0 (pure CMC), the Ra increments of S1, S2, and S3 films were 2.37, 1.24, and 0.57, respectively. The reason might be that the entanglement enhanced with the increase in CTE contents; however, the increments were not linear, which can be supported by the results of Figure 2, Figure 3 and Figure 4. Therefore, the roughness of the film increased as the CET increased.

Figure 8a−d shows the SEM images of the top surface of S0, S1, S2, and S3 films, respectively. It can be seen from the figure that the top surfaces of all films were relatively flat. There were no obvious holes, indicating that CET is uniformly distributed in the CMC matrix, but the degree of uniformity of the distribution was related to the CET content. Moreover, there were tiny cracks in the S0 and S1 films, but the cracks disappeared in the S2 and S3 films. The entanglement made cracks disappear on the top surface of the S2 and S3 films.

Figure 8e−h shows the SEM images of the cross section of S0, S1, S2, and S3 films, respectively. In Figure 8e−h, the tight sheet structure was observed for the S0 and S1 films. Moreover, as the CET content increased, sheet structures interpenetrated each other to show a tight fiber aggregate structure for S2 and S3 films. This suggested that blend films inherited the neatly arranged sheet structure in the CMC film, and the sheet structure in the blend films was neater and extended in the Y axis. Among blend films, the S2 film showed a more uniform sheet structure, while the thickness of the sheet in S3 film appeared more uneven. The uneven sheet thickness in Figure 8h can be explained by the nonuniform distribution of interactions between CMC and CET macromolecules mentioned in Figure 5.

#### 3.2.2. The Relationship between the Morphological Structure of Blend Films and the Rheology of Film-Forming Solution

Figure 9a shows the excitation spectra of the S1–S3 films. The excitation spectra of the S1~S3 film measured at a wavelength of 545 nm peaked at 353, 371 and 380 nm, respectively, showing the ^4^f_8_→^4^f_75_d_1_ transition peak of Tb(III).

Figure 9b shows the emission spectrum of the S1~S3 film measured at an excitation light wavelength of 374 nm. When the excitation wavelength was 374 nm, the S1~S3 film exhibited an emission peak at 488, 543, 590 and 615 nm. In addition, the strongest fluorescence intensity was exhibited by the S2 film, followed by the S3 film and then the S1 film.

The emission peak at 543 nm had the highest fluorescence intensity, assigned to the characteristic ^5^D_4_→^7^F_5_ transition of Tb(III). The emission peak at 488 nm belonged to the ^5^D_4_→^7^F_6_ transition of Tb(III). The emission peak at 590 nm was formed by the superposition of ^5^D_4_→^7^F_4_ transition of Tb(III) and ^5^D_0_→^7^F_1_ magnetic dipole transition of Eu(III). The emission peak at 615 nm was formed by the superposition of the ^5^D_4_→^7^F_3_ transition of Tb(III) and ^5^D_0_→^7^F_2_ electric dipole transition of Eu(III).

Moreover, the fluorescent intensities of the S1 and S2 films increased with the increase in the CET content but decreased in the fluorescent intensity of S3. From the analysis in Figure 2, Figure 3, Figure 4 and Figure 5, the reason for this phenomenon might be the nonuniform aggregation that was caused by the concentration quenching [35,36].

Table 3 shows the effect of CET content on the tensile stress–strain curve of all of the blend films. Compared with the S0 film, the tensile strength of the blend films was improved. Especially, the S2 film was significantly improved to reach the maximum value of 78.76 MPa (see Table 3). The tensile modulus of the blend film increased with the addition of CET, and the blend film with 0.3 g of CET had the largest value. The elongation at the break decreased overall with the increase in the CET content. As compared to other blend films (see Table 4), the blend films reported in this paper had excellent mechanical properties. It was worth noting that the increase in tensile strength was often accompanied by a decrease in elongation at the break, while when the CET content was increased from 0.1 to 0.2 g, the tensile strength of the film was significantly improved, but there was no significant change in elongation at break.

The tensile moduli of the blend films increased with the strength in the entanglements, while tensile strengths and elongation at the break of the S3 film decreased because of the nonuniform aggregation in this S3 film.

Based on the analysis of the structures and properties of the blend films, we can infer that there were interactions between the CMC chain and the CET macromolecule in the blend films to improve the structure and properties of the films. For example, the disappearance of cracks was observed in S2 and S3 films, and a tight fiber aggregate structure was observed in Figure 8. However, as the CET content increased, these interactions had complex effects on the structure and properties of the films rather than a simple promotion. For example, the fluorescence intensity and the tensile strength of the S3 film decreased at the maximum content of CET, as shown in Figure 9 and Table 3 Rheology was involved in investigating the detail of these interactions to illuminate the mechanism of influence on the structure and properties of the films, which can be supported by the results from Figure 2, Figure 3, Figure 4 and Figure 5.

## 4. Conclusions

The AFM micrograph showed the average increase in the roughness of blend films with the increase in the CET amount. The SEM image showed that the cracks on the top surfaces disappeared in the S2 and S3 films. The fluorescence spectra showed that the strongest fluorescence intensity was exhibited by the S2 film, which is followed by the S3 film and then the S1 film. The mechanical test results showed that the S2 film had the strongest tensile strength of 78.76 MPa. Based on the above observations, we assumed the existence of the interactions between the CMC chain and the CET macromolecule in the blend films. Through the Han plot, we confirmed our hypothesis that there were indeed some interactions. The XRD results further confirmed that the interactions were not hydrogen bonds but random entanglements during the drying process. It could be known from the degree of the increment in the roughness of the blend films that the entanglements did not increase linearly with the increased CET content. Moreover, the relaxation time spectrum revealed that the distributions of the interactions between the CMC and CET macromolecules in the S3 film were more nonuniform, which was caused by nonuniform aggregation.

The distinct features of traditional films are responsible for the interest in using them for separation processes. During the past several decades, cellulose-based films have already been dramatically utilized in food, pharmaceutical, and medicine fields, etc. Due to inheriting some unique structures and characterizations, cellulose-based materials with inorganic compounds are exerted in biological systems and advanced applications successfully [36,40,41,42]. In this study, the fabrication of CMC incorporation of cellulose–based nanocomposite can give rise to synergistic functions, that is, the blend films not only have good mechanics properties but also develop fluorescent functionality with narrow emission profiles of rare earth metal ions. This enables potential applications for the films such as probes, sensors and labels, anti-counterfeit technology and monitoring of drug release.

## Data Availability

Not applicable.

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
