# Peer review of "Analysis of Correlation between Structure and Properties of Carboxymethyl Cellulose Film Loaded with Eu3+ and Tb3+ Fluorescence by Rheology at Different Drying Stages"

_polymers, 2022, doi:10.3390/polym14091655_

Round 1

Reviewer 1 Report

  • The abstract must be included with methodology first and then a summary of important results. Please add the methodology briefly in the abstract.
  • What is /Tb-Eu in the abstract? Write the whole name for the first time in the manuscript.
  • Before using the chemicals, did the authors do any purification for chemicals?
  • Section 2.2.1 is incomplete and just is showing the number of materials. Please write the temperature and experiment conditions.
  • Section 2.2.2, please correct the writing style of “10 cm*10 cm”.
  • This sentence in the methodology is wrong: “According to the above steps, determine the drying kinetics curves;”
  • The results and discussions must be compared with other relevant studies.
  • It can be a good idea if the authors add a future and prospects section at the end of the manuscript.
  • The authors can use the following references in this manuscript:

Sabbagh, F., Muhamad, I. I., PaLe, N., & Hashim, Z. (2018). Strategies in Improving Properties of Cellulose-Based Hydrogels for Smart Applications. Cellulose-Based Superabsorbent Hydrogels. Springer International Publishing, 887-908.

Sabbagh, F., & Muhamad, I. I. (2017). Physical and chemical characterisation of acrylamide-based hydrogels, Aam, Aam/NaCMC and Aam/NaCMC/MgO. Journal of Inorganic and Organometallic Polymers and Materials27(5), 1439-1449.

Author Response

Response 1

Q1: The abstract must be included with methodology first and then a summary of important results. Please add the methodology briefly in the abstract.

A1: According to the comment, we rewrote the part of the abstract. (Page1, part of the abstract)

Q2: What is /Tb-Eu in the abstract? Write the whole name for the first time in the manuscript.

A2: We wrote the CMC /europium(Ⅲ)-terbium(Ⅲ) (CET) in the abstract of the revised version. (Page1)

Q3: Before using the chemicals, did the authors do any purification for chemicals?

A3: No, all reagents were used without further purification. We wrote it in our revised version. (Page 2)

Q4: Section 2.2.1 is incomplete and just is showing the number of materials. Please write the temperature and experiment conditions.

A4: We appreciate the reviewer’s suggestion. We rewrote the paragraph in revised version. (Page 2)

Q5: Section 2.2.2, please correct the writing style of “10 cm*10 cm”.

A5: We corrected it as 10 cm×10 cm.(Page 2)

Q6: This sentence in the methodology is wrong: “According to the above steps, determine the drying kinetics curves;” P3 2.2.3

A6: We corrected the sentence in the revised version. (Page 3)

Q7: The results and discussions must be compared with other relevant studies.

A7: We appreciate the reviewer’s suggestion. Table5 displayed that the mechanical properties of our work compared with the other three relevant studies. In order to read clearly, we rewrote the legend. (Page 15,)

Q8: It can be a good idea if the authors add a future and prospects section at the end of the manuscript.

A8: We appreciate the reviewer’s suggestion. A prospect for future applications of our film was shown in the conclusion section.  (Page17)

Q9: The authors can use the following references in this manuscript:

Sabbagh, F., Muhamad, I. I., PaLe, N., & Hashim, Z. (2018). Strategies in Improving Properties of Cellulose-Based Hydrogels for Smart Applications. Cellulose-Based Superabsorbent Hydrogels. Springer International Publishing, 887-908.

Sabbagh, F., & Muhamad, I. I. (2017). Physical and chemical characterization of acrylamide-based hydrogels, Aam, Aam/NaCMC, and Aam/NaCMC/MgO. Journal of Inorganic and Organometallic Polymers and Materials, 27(5), 1439-1449.

A9: These references were cited in the conclusion section. (Page17 & ref. 43 and 44)

Q10: Moderate English changes required

A10: In order to avoid grammatical mistakes and inaccurate language use, we had Dr. Zhouyang Xiang given as a language instructor. He obtained his Ph.D. degree from the University of Wisconsin-Madison in 2015, and he obtained his Master’s degree from North Carolina State University. He published his research works on ACS Applied Materials &Interfaces、Green Chemistry、Carbohydrate Polymers, etc.

   Language had been carefully corrected in the revised version.

Reviewer 2 Report

Having read thoroughly the manuscript, I have suggestive comments as follows:

  1. The title needs to be changed since CMC film doesn’t exhibit fluorescent properties, but Europium (Eu) and Terbium do. Suggesting title may be “ CMC film loaded with Eu2+ and Tb3+ fluorescence.
  2. CTE is a confusing short form in this case since it may be misunderstood with the coefficient of thermal expansion value (CTE) of film. To be more clearly, CTE films should be used in this case.
  3. There are two words found “blend film” and composite. However, in the experiment, there is only a single polymer matrix (CMC) and a crosslinker ( Eu2+ and Tb3+). No experimental support the reinforcing particle in order to claim that this is a composite film. So, I recommend to omit these words from the manuscript.
  4. From my understanding, Eu2+ and Tb3+ act as a crosslinker among CMC chains (CMC-COO--Eu2+--OOC-CMC) (electrostatic bonding) which causes phase separation to occur. I recommend to re-draw Fig 7 in order to convince readers to perceive your work clearly.
  5. Concerning to study continuous film formation (which involves two types of mechanisms, 1: solvent evaporation (your case) and 2: coalescence (latex), the minimum film forming temperature (MFFT) tester must be carried-out to obtain MFFT value. Just a curious that why authors selected 50 oC as a drying time without an explanation. Is it more interesting to study at room temperature to study the relaxation process (in this case).
  6. In the rheological study, authors spent a lot of explanation which might loose a reading interest from audience. I want also want to point-out the particular wordings used in this section “ CMC segment and unreacted CMC chain segment” which correspond to free CMC segment and crosslinked CMC fraction, respectively (in my point of view). Is that the real meaning or not?
  7. What do authors mean by wording “zero shear complex viscosity”? Is it the same as intrinsic viscosity?
  8. Overall, I recommend to make the manuscript more concise and avoid redundancy to attract readers.

Author Response

Response 2

Q1: The title needs to be changed since CMC film doesn’t exhibit fluorescent properties, but Europium (Eu) and Terbium do. Suggesting title may be “ CMC film loaded with Eu2+ and Tb3+ fluorescence.

A1: We rewrote the title according to the reviewer’s suggestion. (Page 1)

Q2: CTE is a confusing short form in this case since it may be misunderstood with the coefficient of thermal expansion value (CTE) of film. To be more clearly, CTE films should be used in this case.

A2: We changed the CTE to CET in the whole revised version.

Q3: There are two words found “blend film” and composite. However, in the experiment, there is only a single polymer matrix (CMC) and a crosslinker ( Eu2+ and Tb3+). No experimental support the reinforcing particle in order to claim that this is a composite film. So, I recommend to omit these words from the manuscript.

A3: We appreciate the reviewer’s suggestion. In our work, we synthesized the CMC/ europium(Ⅲ)-terbium(Ⅲ) composite (CET), according to our previous work. Then we casted the blend film with the mixture of CMC and CET to endow the films with fluorescence property. We corrected the mistakes on P15 (L439-L440, Table 5). Then we showed the Scheme to display the process of film-casting and the structure of CET cited in our previous work. And we addressed that CMC bonded with Eu3+ and Tb3+ both ionically and covalently. (Page 3)

Q4: From my understanding, Eu2+ and Tb3+ act as a crosslinker among CMC chains (CMC-COO--Eu2+--OOC-CMC) (electrostatic bonding) which causes phase separation to occur. I recommend to re-draw Fig 7 in order to convince readers to perceive your work clearly.

A4: Indeed, in our previous work, CMC was crosslinked with Eu3+ and Tb3+ by ionic bond, but also Eu3+ and Tb3+ coordinated with O atoms on CMC chains [1-3]. And we give the Scheme to display the structure of CET. In order to convince readers to perceive our work clearly, we revised the caption and note. (Page 13)

Ref:

  1. Lai, Z.; Ye, .; Xiong, J. Energy transfer processes and structure of carboxymethyl cellulose-Tb/Eu nanocomplexes with color-tunable photoluminescence. 2021, 271
  2. Li, Q.Y.; Ye, J.; Xiong, J.; Abidi, N. Structures and High Fluorescence of Novel Nanocomposites of Sodium Carboxymethyl Cellulose/Tb(III) Prepared at Different pHs. Polym Composite 2017, 38, E498-E507, https://doi.org/10.1002/pc.24013
  3. Ye, J.; Wang, B.; Xiong, J.; Sun, R. Enhanced fluorescence and structural characteristics of carboxymethyl cellulose/Eu(III) nano-complex: Influence of reaction time. Carbohyd Polym 2016, 135, 57-63, https://doi.org/10.1016/j.carbpol.2015.08.063

Q5: Concerning to study continuous film formation (which involves two types of mechanisms, 1: solvent evaporation (your case) and 2: coalescence (latex), the minimum film forming temperature (MFFT) tester must be carried-out to obtain MFFT value. Just a curious that why authors selected 50 ℃ as a drying time without an explanation..

A5: Actually, we investigated the relationship of properties and dry temperatures, such as mechanical properties showing as follows:

Figure effect of drying temperature on mechanical properties of S2

The MFFTs of the films were a little different from CET contents in our study. At around 45℃,their MFFTs, , the film’s formation would take a long time. But over 55℃, the films became cracked or even a piece of chips. So, in this study, we planned to investigate the effect of the CET contents on the structures and properties, then we decided to fix it at 50 ℃ to carry on this study.

Q6: Is it more interesting to study at room temperature to study the relaxation process (in this case)

A6: no. All rheology study was carried out at 50℃ as described in 2.2.3 characterization, in this study.

Q7: In the rheological study, authors spent a lot of explanation which might loose a reading interest from audience.

A7: We appreciate the reviewer’s suggestion. We revised 3.1.1. And we deleted Figure 6 and Figure 10, and rewrote 3.1.5 to make the manuscript more concise and comprehensive.

Q8: I want also want to point-out the particular wordings used in this section “ CMC segment and unreacted CMC chain segment” which correspond to free CMC segment and crosslinked CMC fraction, respectively (in my point of view). Is that the real meaning or not?

A8: We might express unclearly. We use two kinds of materials to cast the films, one is CMC another is CET. When we talk about the interaction between CMC and CET, free CMC segments could confuse readers, because they could refer CMC chains or CET chains which do not react with Eu3+ and Tb3+. However, we think the reviewer’s suggestion makes sense, we revised the term as “CMC segments and unboned CET chain segments”.

Q9: What do authors mean by wording “zero shear complex viscosity”? Is it the same as intrinsic viscosity?

A9: According to polymer physics [1], the intrinsic viscosity is linear in concentration and the quadratic term includes the Huggins coefficient kH in extremely dilute solution. However, in our study, the minimum concentration of film-forming solutions was 13.33mg/mL, we prefer using the term zero shear complex viscosity. According to polymer physics [1], the shear viscosity has defined the ratio of shear stress and shear rate. The zero shear complex viscosity was calculated by Cross model.   

[1] Rubinstein M, & Colby R. Polymer Physics. Oxford university Press. 2003

Q10: Overall, I recommend to make the manuscript more concise and avoid redundancy to attract readers.

A10: Thanks for your kind suggestions. We have revised the manuscript accordingly. For example, we deleted Figure 6, and rewrote 3.1.5 to make the manuscript more concise and comprehensive.

Dr. Zhouyang Xiang helped us improve the manuscript technically. He obtained his Ph.D. degree from University of Wisconsin-Madison in 2015, and he obtained his Master’s degree in North Carolina State University. He published his research works on ACS Applied Materials &Interfaces、Green Chemistry、Carbohydrate Polymers, etc.

       Language had been carefully corrected in the revised version.

Round 2

Reviewer 2 Report

I feel that the manuscript is deserved for publication after making a revision